POINT OF VIEW

# An annotated introductory reading list for neurodiversity

**Abstract** Since its inception, the concept of neurodiversity has been defined in a number of different ways, which can cause confusion among those hoping to educate themselves about the topic. Learning about neurodiversity can also be challenging because there is a lack of well-curated, appropriately contextualized information on the topic. To address such barriers, we present an annotated reading list that was developed collaboratively by a neurodiverse group of researchers. The nine themes covered in the reading list are: the history of neurodiversity; ways of thinking about neurodiversity; the importance of lived experience; a neurodiversity paradigm for autism science; beyond deficit views of ADHD; expanding the scope of neurodiversity; anti-ableism; the need for robust theory and methods; and integration with open and participatory work. We hope this resource can support readers in understanding some of the key ideas and topics within neurodiversity, and that it can further orient researchers towards more rigorous, destigmatizing, accessible, and inclusive scientific practices.

**MIRELA ZANEVA\*, TAO COLL-MARTÍN[†], YSEULT HÉJJA-BRICHARD[†], TAMARA KALANDADZE[†], ANDREA KIS[†], ALICJA KOPERSKA[†], MARIE ADRIENNE ROBLES MANALILI[†], ADRIEN MATHY[†], CHRISTOPHER J GRAHAM[‡], ANNA HOLLIS[‡], ROBERT M ROSS[‡], SIU KIT YEUNG[‡], VERONICA ALLEN[§], FLAVIO AZEVEDO[§], EMILY FRIEDEL[§], STEPHANIE FULLER[§], VAITSA GIANNOULI[§], BILJANA GJONESKA[§], HELENA HARTMANN[§], MAX KORBMACHER[§], MAHMOUD M ELSHERIF[#], ALYSSA HILLARY ZISK[#]**

**\*For correspondence:**
mirela.zaneva@chch.ox.ac.uk

[†]These authors contributed equally to this work
[‡]These authors also contributed equally to this work
[§]These authors also contributed equally to this work
[#]These authors also contributed equally to this work

**Competing interest:** The authors declare that no competing interests exist.

## Introduction

The neurodiversity paradigm can be seen as a shift in thinking that embraces the diversity of minds, brains, and neurocognition, and affirms variation as natural and valuable (*Pellicano and den Houting, 2022*; *Walker, 2021*). Like other paradigm shifts, this change in perspective can be challenging to fully understand in the context of education, research, and social thinking. Here, the issue is further compounded as the term 'neurodiversity' has broad scope. Numerous definitions of neurodiversity as a movement, a research field, and a framework or paradigm exist (*Dwyer, 2022*; *Milton et al., 2020*). These three aspects can be distinguished in the following way: (i) the neurodiversity movement encompasses social, advocacy, and political movements advocating for the rights, inclusion, and acceptance of neurodivergent people; (ii) the neurodiversity research field is a largely academic field studying, for instance, psychological and social aspects of neurodiversity; (iii) the neurodiversity paradigm

or framework is a conceptual framework that at its core challenges medical or deficit-based views of neurodiversity, and instead asserts that neurocognitive differences should be seen as natural variations (see *Table 1*). Moreover, these three aspects can overlap and intersect: for instance, research and activism can intersect in areas such as disability rights, mental health advocacy, social justice, and equity, diversity and inclusion efforts in education and in the workforce (*Clouder et al., 2020*; *Dwyer, 2022*; *Manalili et al., 2023*). Unless specifically discussing one of these aspects, here we generally adopt the term 'neurodiversity paradigm' as a broader idea, encompassing ways of thinking applicable to both the neurodiversity movement and the research field.

Building on this pluralistic perspective, it is important to note that neurodiversity is not limited to cognitive differences, nor to specific named neurotypes (i.e., commonalities in neurological makeup and functioning; *Bottema-Beutel et al., 2021*), though it does include named neurotypes like autism, attention deficit/hyperactivity

disorder (also known as attention dysregulation hyperactivity development [**Dwyer et al., 2024**] or variable attention stimulus trait [**Hallowell and Ratey, 2022**]), and dyslexia, among others. Neurodivergent people typically exhibit neurocognitive variations outside the perceived norm (**Walker, 2021**). However, defining and interpreting neurodiversity remains complex. On the one hand, neurodiversity is viewed through a theoretical lens as a social ecology of mental functions (**Chapman, 2021**). On the other hand, researchers compare neurodiversity to biodiversity in nature (**Silberman, 2015**). Further, people may consider neurodiversity to be a political label, as opposed to a biological label (**Chapman, 2021**; **Ne'eman and Pellicano, 2022**) or conversely, a biological impairment as opposed to 'normal' or neurotypical behavior. Nevertheless, both arguments could undermine neurodivergent people, as neurodivergence can thus be seen as a fictitious identity or a condition defined only by limitations, overshadowing the unique traits of individuals. The debate continues to be contentious, and various definitions have been proposed and debated.

In 2015 Kassiane Asasumasu coined the term neurodivergent and defined it as 'neurologically divergent from typical' (**Asasumasu, 2015**): this definition was broad and inclusive, capturing *all* such forms of divergence, explicitly noting autism, epilepsy, post-traumatic stress disorder (PTSD), cluster headaches, Chiari malformation, attention deficit/hyperactivity disorder (ADHD), multiple sclerosis, Parkinson's, apraxia, cerebral palsy, dyspraxia, various mental health conditions, and neurological differences for which no formal diagnosis has been defined (e.g., aphantasia). Other complementary work has similarly proposed or considered broad views including, for example, dementia (**Silberman, 2023**), as well as mental health conditions like depression and anxiety (**Mellifont, 2019**). However, other have insisted that neurodivergence primarily encompasses neurodevelopmental disabilities like autism, dyslexia, and dyspraxia (**Walker, 2021**) or limit consideration to neurocognitive functions (**Shah et al., 2022**), even while aware of Asasumasu's intentions (e.g., **Monzel et al., 2023**). This lack of consensus, driven by differing theoretical lenses and contexts, underscores the complexity of defining and interpreting neurodivergence within a social construct.

Another challenge for researchers and educators wishing to learn more about neurodiversity, and to implement neurodiversity-affirming practices, is a lack of reliable information about the topic, the presence of misinformation, and the persistence of misunderstandings about neurodiversity (**den Houting, 2019**). Research studies may employ the rhetoric of the neurodiversity movement without a full understanding of its key assumptions (**Neumeier, 2018**), perhaps in part due to lack of well-curated accessible resources. Additionally, researchers might mistakenly believe that the neurodiversity movement only applies to neurodivergent people with lower support needs (often referred to as 'high-functioning'), excluding those they consider 'severe', 'profound', 'high support needs' or 'low functioning' as 'too disabled' (**Jaarsma and Welin, 2012**; note that uncritical use of some of these terms has also been critiqued as ableist within the neurodiversity movement; **Bottema-Beutel et al., 2021**; **Natri et al., 2023**). This can result in the exclusion of neurodivergent people from discussions, despite their valuable perspectives (see **Silberman, 2023** for an argument on how neurodiversity promotes listening). Such exclusion may stem from the assumption that certain groups lack the capacity for self-advocacy or are not given the opportunities needed to be heard. Alternatively, exclusion can result from a kind of disqualification through presumption of low support needs on the basis that neurodivergent people do not face barriers to make their opinions and needs known (**Montgomery, 2005**). In some cases, the same people have experienced exclusion or invalidation both for being presumed too disabled and for being presumed not disabled enough (**Montgomery, 2001**; **Baggs, 2005**). These challenges, combined with limited awareness of diverse neurodivergent groups and a lack of knowledge on implementation strategies, hinder the necessary identification and adoption of inclusive, robust practices in the behavioural, cognitive, and social sciences, as well as in educational and clinical work.

In order to foster interest in neurodiversity initiatives, as well as promote more robust research in the field, an understanding of key ideas and debates, how they have developed, and current perspectives is needed. To facilitate this, an accessible overview introducing key concepts about neurodiversity is crucial. Such an overview should move the field forward and ensure that neurodiversity is promoted and further develops as a paradigm (e.g., **Crüwell et al., 2019**; **Kathawalla et al., 2021**; **Kalandadze and Hart, 2024**). To this end, we have created an introductory reading list. We developed this list collaboratively amongst a community of neurodivergent and neurotypical researchers, guided both by

**Table 1.** Definitions of key terms.

For further context, discussion and examples of these terms, see *Dwyer, 2022* and *Walker, 2021*.

| Term | Definition | Further notes |
| --- | --- | --- |
| Neurodiversity as a paradigm or framework | A conceptual framework for understanding disability that emphasizes the diversity of neurocognitive, social, behavioural experiences and characteristics. | The neurodiversity paradigm challenges deficit-based views of disability by promoting the idea that neurocognitive differences are part of natural human variation, not inherently problematic (*Pellicano and den Houting, 2022*; *Dwyer, 2022*). |
| Neurodiversity as a research field | A broad area of research that encompasses various topics related to neurodiversity, such as investigating the psychological and social aspects of neurodiversity. | The neurodiversity research field can include research on neurodivergent traits across the lifespan, neurodiversity-informed education and workplace practices, among others. For more about neurodiversity approaches for researchers, see *Dwyer, 2022*. |
| Neurodiversity as a movement | A social and political movement that advocates for the acceptance and inclusion of individuals with neurodivergent differences. | The neurodiversity movement seeks to shift public perceptions of neurodivergent people away from seeing them as "disordered" and toward embracing them as part of the diversity of human experiences (*Dwyer, 2022*). |
| Neurodiverse people | A group is considered neurodiverse if its members differ between each other in terms of their neurocognitive functioning. | A neurodiverse group of people can include a mix of individuals who are neurodivergent and individuals who are neurotypical (*Walker, 2021*). |
| Neurodivergent people | Individuals whose neurocognitive functioning diverges from the dominant societal standards of "typical" or "average" functioning. Neurodivergent or neurodivergence may be abbreviated as ND. | Neurodivergent people are those whose experiences diverge from what is considered neurologically typical. Asasumasu coined this term with the intention of covering all forms of divergence, including autism, ADHD, epilepsy, cluster headaches, among others (*Asasumasu, 2015*) |
| Neurotype | A term used to describe a particular common pattern of neurocognitive functioning. | Examples of some named neurotypes include autism, ADHD, dyslexia. See *Bottema-Beutel et al., 2021* for more on avoiding ableist language. |

research expertise and lived experience. We first made an open call for reading recommendations on neurodiversity, targeting thoughtful and impactful literature. The open call was primarily shared through the Framework for Open and Reproducible Research Training (FORRT) platform, an online collaborative community focused on promoting open scholarship, as well as advancing research and education about neurodiversity. Recommendations were welcomed from both academic and non-academic sources, with no exclusion criteria for specific topics or formats. Contributors were asked to explain the strengths of their suggestions, and each recommendation was independently verified by a second researcher. Three of the authors then analysed the papers and categorised them into themes, and the final list of papers and themes was refined through further discussions. The selection process aimed to prioritize materials that were educational, thought-provoking, and broadly relevant (see Methods for more details). In the end we agreed to have nine themes, with two articles for each theme (see appendix 1 for a list of the 18 articles). The articles also vary regarding their own use of language, viewpoints regarding neurodiversity, and positionality (see *Box 1*).

The first six themes – the history of neurodiversity; ways of thinking about neurodiversity; the importance of lived experience; a neurodiversity paradigm for autism science; beyond deficit views of ADHD; and expanding the scope of neurodiversity (by, for example, including dyslexia, dyspraxia, developmental language disorder, and stuttering) – are intended to provide a fundamental understanding of neurodiversity. Rather than being completely discrete and self-contained, these themes often cover related topics, albeit from different perspectives and with different examples. For instance, commonalities emerge regarding the importance of neurodiversity affirming models over deficit-based views, the use of appropriate language, the need to expand definitions of neurodiversity (for instance, to be inclusive of mental health), and the need for inclusion and centring of lived experience. The final three themes – anti-ableism; the need for robust theory and methods; and integration with open and participatory work – are different in that they are oriented towards the future and include actionable steps for future work.

Overall, we hope that this reading list will be a resource that can support readers in obtaining a fundamental and holistic understanding of neurodiversity, and that it will also encourage

## Box 1. Notes on language and ideas captured in the reading list.

The articles in our reading list are varied in terms of their topics, publication time and cultural context, so they vary regarding their use of language, viewpoints regarding neurodiversity and positionality (see Methods for our own positionality statement). On the whole, we have strived to highlight important and productive ideas about neurodiversity, while rejecting stigmatizing and ableist views.

Our collective social and research understanding of how stigma and ableism work advances over time, so we therefore wish to acknowledge that research standards and views around what constitutes (in)appropriate positionality and language also change. This is especially important in the context of neurodiversity; as neurodiversity is not a 'monolith', different areas of study or social activism have their own current standards. In the present paper, we have leveraged both the research expertise and lived experiences in our team to come to a general agreement about how to highlight important work, while minimizing harm. We have done this both by open general discussion, where all members from the team were welcome to feedback on all papers, at any time, as well as by more targeted reading. Specifically, all highlighted papers were independently read by at least three people (most by five, all of these independent readers were not involved in recommending, double-checking or summarizing the corresponding papers) to ensure that at a broad level, the core ideas were not stigmatizing, ableist, or harmful. Further to this end, for the more widely studied topics within neurodiversity, such as autism and ADHD, we also required that academic papers do not consider autism or ADHD through an exclusive deficit-based view (for instance, treating autism or ADHD as disorders or separating individuals into 'high' or 'low' functioning based on arbitrary statistical cut-offs). We did not impose such restrictions on language for areas of neurodiversity that have been historically understudied, such as Developmental Language Disorder, stuttering or dyspraxia, where we worried that further exclusion of these bodies of work may decrease their recognition as important fields within neurodiversity. Lastly, we acknowledge that despite our quality assurance procedures, it is possible that some of the (hopefully more granular) ideas expressed within the selected papers can still be controversial and debated – for instance, we note in passing, that some papers included brief generalizing statements or phrases that could be negatively charged (e.g., communication impairments instead of communication differences). In this regard, the fact that we have highlighted a certain paper does not mean we agree with all of its ideas or language used. We nevertheless strived to only include papers if their core ideas were, in our joint opinion, not stigmatizing or ableist.

researchers to apply more rigorous and destigmatizing scientific practices.

### Themes

#### *History of neurodiversity*

Recommendation 1: Botha M, Chapman R, Giwa Onaiwu M, Kapp SK, Stannard Ashley A, Walker N. 2024. The neurodiversity concept was developed collectively: An overdue correction on the origins of neurodiversity theory. *Autism* **28**:1591–1594.

Recommendation 2: Sinclair J. 1993. Don't mourn for us. *Autism Network International: Our Voice Newsletter* Issue 3; pages 20–22.

The neurodiversity movement emerged in the 1990s, following the influences of the autistic rights movement and earlier disability rights movements of the 1960s and 70s (*Botha et al., 2024*; *Kapp, 2020*). Many have tried to pinpoint the exact moment when the term 'neurodiversity' emerged. **Botha et al**. refer to recent archival examinations of extant texts from the 1990s, including forums, community email lists such as Independent Living (autism community), and records of community members and prominent activists of the time, including Tony Langdon in 1996 and Harvey Blume in 1997 and 1998. The authors highlight that many people throughout the 1990s discussed ideas about 'neurological

diversity', with the specific term 'neurodivergent' later coined in the 2000s by Kassiane Asasumasu (*Asasumasu, 2015*). Considering this, Botha et al. argue that the idea of neurodiversity was collectively developed. This corrects a common erasure of neurodivergent people from their own history in misattributing the term singularly to Judy Singer's first academic use in her 1998 honours thesis and shows the neurodiversity movement has always had a strong community spirit.

For many, neurodivergent communities offer belonging, social connectedness, a way to share experiences and perspectives, and practical support and advice, including empowerment (*Botha et al., 2022*). Empowerment is essential for wellbeing, self-efficacy, and acceptance, especially for neurodivergent people who face greater risks for isolation, stigmatization, negative stereotyping, and even victimization, with a recent meta-analysis showing that almost half of autistic people had experienced some form of victimization (*Trundle et al., 2023*). The early autistic self-advocacy movement of the 1990s was acutely aware of these risks faced by autistic and broader neurodivergent communities (for a critique of early behavioural interventions, see *Yergeau, 2018*). One salient response can be found in **Jim Sinclair's speech** 'Don't Mourn for Us' presented at the 1993 International Conference on Autism in Toronto (*Sinclair, 1993*). This speech, primarily directed at parents of autistic children, underscores the importance of understanding autism – and indeed neurodiversity – not through a focus on perceived deficits, but by appreciating each person in their own right. These ideas still form part of critical debates around whether and how intervention practices could align with inclusive, participatory, and non-stigmatizing approaches to fostering neurodivergent wellbeing (*Leadbitter et al., 2021*).

### How do we think about neurodiversity?

Recommendation 1: Dwyer P. 2022. The neurodiversity approach(es): What are they and what do they mean for researchers? *Human Development* **66**:73–92.

Recommendation 2: Constantino CD. 2018. What can stutterers learn from the neurodiversity movement? *Seminars in Speech and Language* **39**:382–396.

Collective understandings of neurodiversity have evolved significantly in the last 30 years and it can be challenging to trace back and understand this evolution without context. Two key papers examine the history of neurodiversity and its key

ideas (*Dwyer, 2022*; *Constantino, 2018*). Both analyze how medical, social, and contemporary models of neurodiversity offer different tangible targets for research. Researchers, activists and laypeople increasingly refer to natural variation in human brains, behavior, and cognition as neurodiversity and consider neurocognitive variants like autism, ADHD, dyslexia, stuttering and others as part of this natural variation rather than only 'disorders' that always need to be 'cured' or 'fixed'. A growing body of socio-environmental research suggests the difficulties neurodivergent people face cannot fully be understood at the individual level, but rather societal barriers and their interactions with personal characteristics, abilities, and circumstances should also be examined. This paves the way for both environmental and societal support, including reasonable adjustments, increased accessibility, anti-discrimination protections, as well as individual-level support (e.g., learning adaptive skills).

Building on these ideas, **Dwyer** recommends researchers interested in neurodiversity do not exclusively focus on studying perceived weaknesses, but instead balance such research with also studying neurodivergent people's strengths and how they can be leveraged to help neurodivergent people thrive and achieve their goals.

In a similar light, **Constantino** argues therapy and interventions should focus on people's wellbeing rather than perceived 'normalization' of particular behaviours. As an illustrative example, this could mean that when providers offer early interventions to young stutterers, the sole focus need not be placed on fluency but could entail assisting young people with their subjective experience of stuttering, affirming their emotions, and helping improve their wellbeing (*Shenker et al., 2023*).

### The importance of lived experience

Recommendation 1: Johnson RM. 2023. Dyslexia is not a gift, but it is not that simple. *Infant and Child Development* **32**:e2454.

Recommendation 2: van Gorp R. 2022. My journey and the value of a community where neurodiversity is celebrated. *Scope Contemporary Research Topics: Learning and Teaching* **11**:42–49.

The neurodiversity movement, with its focus on advocating for neurodivergent people, serves as a framework through which advocates, practitioners, and researchers challenge traditional assumptions about neurodivergent experiences. Prior to the emergence of neurodiversity as a

paradigm, the dominant approach – rooted in biomedical psychiatry – categorized individuals into 'mentally disordered' subgroups based on their symptoms (*Chapman, 2021*; *Hunt and Procyshyn, 2024*). This medical model has been critiqued as dismissive of people's experiences, by treating them as unreliable (and individual/anecdotal), and perhaps even limiting people's opportunities to independently understand their own thoughts, feelings, and behaviours (*Cutler, 2019*; *Petty and Ellis, 2024*). In the context of neurodiversity literature, lived experiences refer to the unique and subjective perceptions, narratives, and encounters of those who identify as neurodivergent. These accounts provide valuable insights into the day-to-day realities, triumphs, and challenges of neurodivergent minds (see *Kidd, 2018* on traumatic brain injury). Through shared experiences, neurodivergent communities might gain empowerment, validation, improved self-efficacy and wellbeing, as well as increased social support, connectedness, and reduced feelings of isolation (*Watts et al., 2024*; see also *Milton et al., 2020* for wider discussion of the value of lived experience).

Lived experiences also benefit researchers studying neurodivergent people. For example, **Johnson** argues that valuing these experiences is crucial for gaining a nuanced understanding of dyslexic perspectives. Researchers should actively seek partnerships with dyslexic people to incorporate their feedback and centre their voices within dyslexia research. Furthermore, neurodivergent researchers themselves can contribute by sharing their personal experiences. Doing so will not only reduce stigma, it will also spread knowledge about coping mechanisms and tools and illuminate the intersections of neurodivergent experiences and professional careers.

**van Gorp** shares her journey navigating educational spaces over time, both as a neurodivergent student and lecturer (*van Gorp, 2022*). She details her experiences with being diagnosed with Irlen syndrome and dyslexia, as well as her decision to disclose her diagnosis at a Neurodiversity Symposium, and the subsequent empowerment and community support she felt. Indeed, both van Gorp and Johnson emphasize that sharing lived experiences fosters empowerment, inclusion, and compassion, ultimately enriching our collective understanding of neurodiversity.

## A neurodiversity paradigm for autism science

Recommendation 1: Pellicano E, den Houting J. 2022. Annual Research Review: Shifting from 'normal science' to neurodiversity in autism science. *Journal of Child Psychology and Psychiatry*. **63**:381–396.

Recommendation 2: Botha M, Hanlon J, Williams GL. 2023. Does language matter? Identity-first versus person-first language use in autism research: A response to Vivanti. *Journal of Autism and Developmental Disorders* **53**:870–878.

The field of autism research has a long history predating the neurodiversity movement, and consequently, both scientific and social understandings of autism have developed over time (*Kapp, 2020*). The two papers highlighted here poignantly argue for the need to move towards a neurodiversity paradigm for autism science (*Pellicano and den Houting, 2022*), and engage more deeply with considerations around language use, particularly by centring the needs, autonomy and rights of autistic people (*Botha et al., 2023*).

In their review article, **Pellicano and den Houting** acknowledge that the conventional medical approach has advanced our understanding of autism, but they also argue that this approach has been challenged due to the rise in autistic self-advocacy, the neurodiversity movement, and the relative absence of non-deficit based explanations regarding what autism is. The authors focus on big-picture ideas related to the neurodiversity paradigm and its vital application to autism science: (i) focusing on relational contexts, systemic contexts, and the interaction between contextual and individual factors rather than attributing all difficulties for all parties to deficits within one (autistic) party; (ii) supporting autistic contributions to autism research, including through support for autistic researchers, collaborations involving autistic people (both lay community members and researchers), and the development of more robust participatory mechanisms for co-design and co-production; and (iii) focusing on autistic community priorities, ensuring research-generated knowledge is translated into real-world applications targeting the challenges autistic people face.

**Botha, Hanlon and Williams** discuss the use of language in autism research, focusing on the priorities of the autistic community. Their work offers a rich treatment of the differences between person-first language and identity-first language, while acknowledging that there is currently no

clear majority consensus among autistic people in terms of preferred language; there is a need to replicate and expand previous survey efforts. Crucially, they argue that language use is highly important, with tangible consequences including stigmatization and dehumanization. With this in mind, research and practice should centre the needs and experiences of autistic people.

### Beyond deficit views of ADHD

Recommendation 1: Sonuga-Barke EJ. 2023. Paradigm 'flipping' to reinvigorate translational science: Outlining a neurodevelopmental science framework from a 'neurodiversity' perspective. *Journal of Child Psychology and Psychiatry* **64**:1405–1408.

Recommendation 2: Tamir T. (2023). Being Neurodivergent in Academia: Working with my brain and not against it *eLife* **12**:e95068.

ADHD is increasingly being explored via a neurodiversity lens through works that have two aims: (i) to provide a rich understanding of ADHD; (ii) to reshape practical applications in everyday and professional environments. One such example is an opinion article in which **Sonuga-Barke** critiques the traditional biomedical model that has long dominated ADHD research and therapy, proposing instead a neurodiversity-affirming model (*Sonuga-Barke, 2023*). In particular, it introduces an innovative intervention program that can be implemented by neurodivergent researchers. In addition to challenging existing ways of thinking, this new approach also actively involves neurodivergent people in the creation and execution of research, thereby ensuring that the interventions are genuinely reflective of and responsive to the needs of those with ADHD.

Another compelling exploration of ADHD is presented by **Tamir**. The article highlights the personal journey of an academic who initially received a diagnosis of depression during their PhD studies (*Tamir, 2023*). Years later, an ADHD diagnosis clarified the root of their ongoing struggles with mental health, spurred in part by the high demands of academia. This narrative underscores the often-misunderstood manifestations of ADHD, such as hyperfocus and impulsivity, which, while sometimes beneficial in a research setting, frequently lead to burnout. In addition to sharing a personal story, the author also discusses strategies that can be adapted to harness ADHD traits beneficially.

Both articles advocate for a shift away from viewing ADHD through a deficit lens to recognizing it as part of the broader spectrum of human neurocognitive diversity. They call for educational and professional systems that do not merely accommodate but actively embrace and adapt to neurodivergent ways of thinking and learning, promoting a more inclusive environment.

### Expanding the scope of neurodiversity: diverse neurotypes and experiences

Recommendation 1: Green AE, Alyssa, Durá L, Harris P, Heilig L, Kirby B, McClintick J, Pfender E, Carrasco R. 2020. Teaching and researching with a mental health diagnosis: Practices and perspectives on academic ableism. *Rhetoric of Health & Medicine* **3**: Issue 2; article 1.

Recommendation 2: Elsherif MM, Wheeldon LR, Frisson S. 2021. Do dyslexia and stuttering share a processing deficit? *Journal of Fluency Disorders* **67**:105827.

Historically, neurodiversity work has focused on autism and ADHD, though our understanding of neurodiversity is broader (*Asasumasu, 2015*) and includes mental health and language-based disabilities, which we highlight here to showcase the diversity of neurodivergences and experiences. **Green et al**. explored the experiences of nine people who navigate their mental health diagnoses within academia (*Green et al., 2020*). Through a dialogue format they discuss various challenges, including around getting a diagnosis, decisions regarding disclosure, managing the limitations and affordances of their disabilities, seeking reasonable adjustments, and advocating for themselves. They also argue that while disability laws in their country acknowledge these needs, those with mental disabilities are still seeking access to education, care, appropriate accommodations, among others. Their work highlights the need to improve inclusivity by promoting conversations about mental health within academic environments.

**Elsherif, Wheeldon and Frisson** assessed the potential language processing link between dyslexia and stuttering through a prevalence study in a British sample of 164 adults (*Elsherif et al., 2021*). They found that 43% of dyslexics stuttered during childhood, and 50% of stutterers were identified as dyslexic. Considering their use of medical model language (e.g., deficit), we can reframe their findings through a neurodiversity-affirming lens: (i) they provide evidence that dyslexia and stuttering co-occur; (ii) they carve paths so future research can rigorously investigate whether dyslexia and stuttering

have similar phonological profiles; (iii) their findings may help dyslexics and stutterers be better understood and supported within academia and society. Such reframings align with the push for inclusivity in research concerning dyslexia, stuttering (*Constantino, 2018*; *Taylor et al., 2023*), and the broader field of speech/language pathology (*Manalili, 2022*). We also caution against oversimplification when studying neurodiversity. Dyslexia, stuttering, and other forms of neurodivergence need not be seen as 'gifts' to be valued; as others have argued, even 'positive' stereotypes could be harmful (*Odegard and Dye, 2024*). Instead, it is important to recognize various forms of neurodivergence inherently as variations that contribute to the richness of neurodiversity (*Johnson, 2023*).

### Anti-ableism

Recommendation 1: Natri HM, Abubakare O, Asasumasu K, Basargekar A, Beaud F, Botha M, Bottema-Beutel K, Brea MR, Brown LXZ, Burr DA, et al. 2023. Anti-ableist language is fully compatible with high-quality autism research: Response to Singer et al.(2023). *Autism Research* **16**:673–676.

Recommendation 2: Hamilton LG, Petty S. 2023. Compassionate pedagogy for neurodiversity in higher education: A conceptual analysis. *Frontiers in Psychology* **14**:1093290.

Anti-ableism and anti-ableist language go far beyond the framework of neurodiversity. Anti-ableism is part of the broader disability rights movement, a social movement against discrimination and bias toward disabled people. Specific forms of ableism include psychophobia or sanism, referring to discrimination against people with mental health problems and who, as a result, are 'psychiatrized' (i.e., caught in the medical world and sometimes locked in psychiatric institutions; *Chamberlin, 1978*). Language can play a role in shaping perceptions and attitudes towards people with disabilities, including those with mental disabilities or other forms of neurodivergence. Many studies on neurodivergence are conducted within an exclusively medical and psychiatric framework, which can sometimes reflect biases (*Bottema-Beutel et al., 2023*). These studies are often carried out by neurotypical researchers, which may inadvertently influence the way neurodivergent individuals are represented. This highlights the importance of adopting more inclusive practices in research, particularly when it comes to language.

However, recommendations for more inclusive and neutral language can often be controversial. While some have argued that more neutral language would hinder scientifically precise descriptions (*Singer et al., 2023*), those promoting anti-ableist language argue ableist terminology is often *both* irrelevant and pejorative. For example, **Natri et al**. propose using 'likelihood' instead of 'risk', and 'co-occurring' instead of 'comorbidity'. Similarly, the terms 'profound autism', 'severe' or 'challenging behavior' can be dehumanizing. Moreover, they are often vague and overly simplistic, as they tend to imply clear-cut divisions on a linear scale of severity (e.g., between 'low' and 'high functioning'). Such terms may also ignore or minimize other important dimensions of a person's experience, including their full range of abilities, interests and needs, as well as how these vary across different contexts (*Zisk, 2019*).

While ableism can be reflected in language, it is not limited to linguistic expression alone, and so anti-ableism efforts should extend beyond language. In the context of anti-ableism in education, **Hamilton and Petty** propose establishing a compassionate educational paradigm that emphasizes empathy, inclusiveness, and care (*Hamilton and Petty, 2023*). The goals of such efforts are to provide more flexibility in how students access course content and demonstrate their learning, as well as to encourage neurodivergent students to build positive schemas for themselves in an educational context.

### The need for robust theory and methods

Recommendation 1: Gernsbacher MA, Yergeau M. 2019. Empirical failures of the claim that autistic people lack a theory of mind. *Archives of Scientific Psychology* **7**:102.

Recommendation 2: Cheng Y, Tekola B, Balasubramanian A, Crane L, Leadbitter K. 2023. Neurodiversity and community-led rights-based movements: Barriers and opportunities for global research partnerships. *Autism* **27**:573–577.

To advance the scientific study of neurodiversity, robust theory and methods are essential. We highlight two papers with useful insights regarding how such efforts can be advanced. **Gernsbacher and Yergeau** critique a large body of work that erroneously claimed that autistic people lack theory of mind, ultimately finding that the evidence base is 'empirically questionable and societally harmful'. They do this by pointing out failures in the literature regarding specificity, universality, replication, convergent validity, and

predictive validity – thus also offering benchmarks of standards that future research should meet. Gernsbacher and Yergeau offer many examples of specific research tasks that were either inappropriate to test for theory of mind, too 'narrow' in focus, or lacking in convergence between each other (e.g., tasks whose results do not correlate). Overall, these papers powerfully illustrate how poor research practices can perpetuate harmful stereotypes and how critical engagement with more rigorous and robust research standards can help to address these problems.

Further important aspects for developing robust methods for studying neurodiversity include asking useful research questions with relevance to neurodivergent people's lives and needs (see next theme), questioning who gets to be included in conversations and work on neurodiversity, and understanding neurodiversity as a global, rather than solely western area of research and activism. Historically, the neurodiversity movement has been driven largely by English-speaking White autistic people, primarily from countries in the Global North, and it is necessary to recognize the issue of intersectionality in terms of those whose voices have been included in neurodiversity activism and scholarship and those whose voices have been excluded (for instance, on the need for greater racial diversity in autism research, see *Giwa Onaiwu, 2020*). Indeed, as some have warned, solidifying the homogenization of neurodiversity as a White and western movement could undermine the social justice and emancipatory goals of the movement (*Nair et al., 2024*).

Drawing on their collective experiences in Ethiopia, India, and Hong Kong, **Cheng et al**. can help readers think critically about the issues of intersectionality and inclusion by discussing a variety of sociocultural and political conditions specific to Asian and African neurodiversity efforts (*Cheng et al., 2023*). Cheng et al. argue that the neurodiversity movement shares fundamental goals with decolonization agendas such as dismantling what, at times, to some may have seemed as 'objective' scientific efforts that ultimately disparage the truths, knowledge, and priorities of lived experiences (e.g., claims that autistic people lack theory of mind). In this light, decolonizing knowledge production, respecting local theoretical frameworks, indigenous knowledge, and fostering community-led science could be important tools for a more robust study of neurodiversity that does not dehumanize neurodivergent people.

## Integration with open and participatory work

Recommendation 1: Gourdon-Kanhukamwe A, Kalandadze T, Yeung SK, Azevedo F, Iley B, Phan JM, Ramji AV, Shaw JJ, Zaneva M, Dokovova M, Hartmann H, Kapp S, Warrington K, FORTT, Elsherif M. 2023. Opening up understanding of neurodiversity: A call for applying participatory and open scholarship practices. *The Cognitive Psychology Bulletin* **8**:23–27.

Recommendation 2: Heraty S, Lautarescu A, Belton D, Boyle A, Cirrincione P, Doherty M, Douglas S, Plas JRD, Van Den Bosch K, Violland P, Tercon J, Ruigrok A, Murphy DGM, Bourgeron T, Chatham C, Loth E, Oakley B, McAlonan GM, Charman T, Puts N, Gallagher L, Jones EJH. 2023. Bridge-building between communities: Imagining the future of biomedical autism research. *Cell* **186**:3747–3752.

From its inception, the neurodiversity movement has advanced through collective action and conversation (see 'History of neurodiversity' above). **Gourdon-Kanhukamwe et al**. consider the power of inclusive collective work as important as ever, with concrete opportunities to catalyse and inspire such efforts within the frameworks of participatory and open scholarship. Large 'big team science' initiatives within the open scholarship movement, such as ABRIR (Advancing Big-team Reproducible science through Increased Representation) and FORRT, have successfully enabled a variety of projects designed by more diverse communities of researchers. The Team Neurodiversity initiative within FORRT, for example, maintains a Database of Neurodivergent Researchers and has provided support for a number of projects on participatory research and open scholarship (*Elsherif et al., 2022*; *Gourdon-Kanhukamwe et al., 2023*; *Phan et al., 2025*). Other groups, such as the Feminist WonderLab (*Hartmann et al., 2024*) or newly emerging NeurodiversiTea journal clubs, strive to make academia a better place for underrepresented people.

To foster productive participatory work with mutual trust and without tokenism, **Heraty et al**. argue that it is important to have purposeful involvement at all stages of the research process, including selecting research questions, designing studies and protocols, and interpreting and disseminating findings (*Heraty et al., 2023*).

Both Heraty et al. and Gourdon-Kanhukamwe et al. highlight many of the benefits of involving neurodivergent people in co-production and mutuality practices of research, including the

promotion of wider epistemic justice, equality in knowledge production, greater relevance of research to lived experience, and greater translational potential of research findings.

### Further readings

We hope the themes discussed here spark an interest in neurodiversity. The current list of themes is not intended as a canonical or definitive organization, and is only one of many possible ways to learn more about neurodiversity. As such, we want to offer suggestions for further key readings, as well as other potential themes or topics of interest. Readers interested in extended introductions about the neurodiversity paradigm, movement, and research field, may wish to consider *The Neurodiversity Reader* (*Milton et al., 2020*). Detailed accounts about the autistic community, its early development, and key figures should consult the book *Autistic Community and the Neurodiversity Movement: Stories from the Frontline* (*Kapp, 2020*). For deeper theoretical engagement with neurodiversity, as well as critiques to medicalized views and harmful societal viewpoints, please see the books *Neuroqueer Heresies* (*Walker, 2021*) and *Authoring Autism: On Rhetoric and Neurological Queerness* (*Yergeau, 2018*).

It is important to recognize the plurality of important topics in neurodiversity and the inherent subjectivity in thematic categorization. For instance, many of the papers we synthesized here could have been thematically organized in different ways, such as focusing more on: the evolution of the neurodiversity paradigm over time; different global and cultural perspectives on neurodiversity; the importance of intersectionality in shaping neurodivergent experiences; better understanding stigma, ableism, and language; charting neurodiversity across the lifespan; and understanding how lived experience can both drive academic research and theory and also be a theoretical contribution. Indeed, as online forums and discussions show, the term neurodiversity was collectively developed (*Botha et al., 2024*; see also 'History of neurodiversity' above). This is only one of many examples of the power of community discussion and community theorizing, and the importance of meaningful engagement with community work (*Zisk, 2024*). With this in mind, we also wish to encourage interested readers to consider blogs. As one example, we recommend *Autistic Scholar* by Patrick Dwyer, a blog with rich discussions informed by both academic research and lived experience on contemporary topics such as the double empathy problem (*Dwyer, 2024*). Blogs can also chart the evolution and spread of ideas, such as an important discussion by Mel Baggs on the history and importance of the concept of neurodivergent 'cousins' (that is, people who share common communication patterns or social characteristics without necessarily sharing the same neurotype; *Baggs, 2016*). We also direct interested readers to a reading list on critical autism studies beyond academia complied by one of the present authors (*Zisk, 2023*).

## Conclusion

This paper aims to serve as an accessible resource for researchers, educators and students to better understand neurodiversity and to support neurodivergent people. It is important to develop neurodiversity, both as a paradigm and social movement, and in rigorous and inclusive ways. Past research, carried out with poor theoretical and methodological approaches, has likely reinforced harmful stereotypes (e.g., erroneous claims that autistic people lack theory of mind; *Gernsbacher and Yergeau, 2019*). The eradication of such harmful stereotyping and discrimination will remain challenging as long as existing barriers, including a lack of awareness and knowledge about neurodiversity and its heterogeneity, persist. To address these challenges, we have curated and presented different key papers that contribute and advance our understanding of neurodiversity. We hope researchers, educators, scholars, activists and neurodiversity allies build on this effort and further promote a positive and productive neurodiversity field.

This reading list focused not only on what neurodiversity is or has been historically (e.g., medical classifications of disorders), but what it can be. We envision a future where everyone is welcomed, valued, and listened to, where weaknesses are acknowledged without pathologization, and strengths are celebrated, leading to continual improvement and positive growth.

## Methods

In August 2023, we published an open call for contributions via the Framework for Open and Reproducible Research Training (FORRT) community channels and personal contacts (available on OSF: https://osf.io/c98sk/). We believe FORRT reaches a fairly diverse audience of people working in and outside of academia, interested in open scholarship and neurodiversity. Some of

our collaborators were already involved in other FORRT projects, whereas others were familiar with FORRT but not involved in this community; some collaborators also joined after hearing about our open call from their own networks. At a broad level, our approach entailed collecting reading recommendations and then double-checking and categorizing all recommended materials.

We were interested in finding thoughtful and robust literature that could provoke discussion, reflection and interest in the field of neurodiversity. We asked people suggesting materials to prioritize articles they were particularly impressed by, that had changed or challenged their thinking, or considered to be fundamental contributions to the field. We did not apply any exclusion criteria regarding the specific field, topic, research method, design, or population studied. We welcomed both empirical (e.g., original research) and theoretical (e.g., position statements) pieces of work. We anticipated that the majority of articles included would be peer-reviewed manuscripts. This was not a formal inclusion criterion, as discussions on neurodiversity also originate outside the academic sphere and continue to be a vibrant topic of conversation beyond formal research settings (*Zisk, 2023*). We aimed to be as inclusive as possible in order to not miss any potentially relevant content (e.g., working papers, viewpoints, newspaper articles, blogs, manifestos, letters and correspondence). This was done with the particular consideration that position statements, co-produced work, or work with embedded mutuality practices may not always be presented in 'traditional' academic formats.

Contributors who submitted reading materials for consideration for the annotated reading list were asked to provide an explanation for their suggestions, describing the strengths or contributions of the specific papers or materials they recommended. People could suggest work that they authored or contributed to. However, to reduce conflict of interest and bias, authors were required to disclose this information. Each paper suggestion was double-checked by a second, independent researcher, who verified citations, content explanations, and optionally provided further comments or personal reflection about the importance of the proposed reading material.

Three of us (MZ, MME, AZ) then examined all papers, as well as the reasons for recommendations, and any comments, and provided a first thematic categorization of all papers. This categorization was discussed with the entire team. After feedback and discussion with all collaborators, the themes were finalized. Then, we discussed as a group which two papers to highlight per theme. Given the existing varying research specialties in our groups, team members who had research expertise, lived experience, and/or interests relevant to each theme volunteered to finalize the selection of papers and draft a statement of the importance of the reading materials for the corresponding theme. We had on average 2–3 volunteers who worked on drafting each theme directly. For each theme, we recommended that 1–2 papers are highlighted but instructed volunteers that they could select more. We originally suggested 1–2 papers as a guiding number, bearing in mind that other annotated reading lists have typically highlighted and annotated one key topic per theme (e.g., *Kalandadze and Hart, 2024*; *Crüwell et al., 2019*). We also worried that selecting many more papers per theme could lead to a potentially overwhelming reading list for readers.

Overall, we prioritized papers that we, as a group, considered were fundamentally important, educational, and thought-provoking. We considered papers to have high educational value if they (for example) provided clear overviews of a given topic, if they covered a debate or traced a given idea or concept's historical origin comprehensively and with appropriate context. In order to consider texts as educational, we took into account how accessible they were for readers, especially potential newcomers to neurodiversity. With this in mind, we wanted texts to be intelligible to a wider audience and not exclusively geared to experts. We were still inclusive of texts with some degree of jargon or specific terminology as long as on the whole, the core messages were clear, and texts were informative. Regarding the importance of texts, we looked out for work that made a clear contribution with relevance to neurodiversity (as a paradigm, a research field, or a social movement). Given the variety of types of texts and topics, importance could relate to different aspects, such as changing discourse, clarifying a concept, challenging a widely held belief, offering a novel perspective, etc. We did not require papers to be scored numerically against a specific rubric or scale (e.g., for importance) but instead used these broad guiding principles (see Appendix 1). Although such criteria can be subjective, we hope that the plurality of research interests and lived experiences in our group have minimized potential individual level biases. Interested readers can find short summaries and comments on the importance of the papers on our OSF

page (https://osf.io/c98sk/) in the reading list and double checking excel sheet forms.

For the purposes of this annotated reading list, we collected and double-checked 54 items. We categorized a final selection of 18 papers, chosen based on their subjective importance, covering nine themes: history of neurodiversity; how do we think about neurodiversity?; the importance of lived experience; a neurodiversity paradigm for autism; beyond deficit views of ADHD; expanding the scope of neurodiversity; anti-ableism; the need for robust methods; and integration with open and participatory work.

## Positionality

We are a diverse group of both neurodivergent and neurotypical researchers, working in and outside of academia in different countries around the world and at different career stages. We are united by our shared interest in neurodiversity on personal and/or scientific levels. Most of our team members form part of the FORRT community. This is an open group for all, where we strive to promote open scholarship, as well as values of social justice, diversity, inclusion, belongingness and equity. The current manuscript was written as a joint, collaborative work, where anyone interested in contributing could do so. The core criteria for authorship entailed suggesting at least two items and checking at least two items. Additional tasks such as theme and paper selection, drafting, editing, analysis, and administrative support contributed to author order, and in a small number of cases substituted for material suggestions and/or checking. This led to five groups of authors of varying size, with equal contributions within each group.

As we come from different academic, professional, educational, and personal backgrounds, and similarly have different degrees of privilege, different abilities and skills in different domains, we hold different views on what constitutes 'neurodiversity' and how it or its different facets should be most appropriately described and positioned. We view this plurality and divergence of viewpoints as positive and productive, allowing a greater inclusion and consideration of varying perspectives. Our core aim with this annotated list is not to be prescriptive about neurodiversity, but rather to introduce readers to important views on critical topics in the field, such as key historical and current trends, as well as open discussion about how to strengthen the field.

**Mirela Zaneva** is at Christ Church College, University of Oxford, Oxford, United Kingdom

mirela.zaneva@chch.ox.ac.uk

https://orcid.org/0000-0003-3569-931X

**Tao Coll-Martín** is at the Mind, Brain, and Behavior Research Center (CIMCYC) and the Department of Behavioral Sciences Methodology, University of Granada, Granada, Spain

https://orcid.org/0000-0002-0591-4018

**Yseult Héjja-Brichard** is at the CNRS, Montpellier, France and the University of Maryland, Baltimore County, Baltimore, United States

https://orcid.org/0000-0003-3939-3852

**Tamara Kalandadze** is at Østfold University College, Halden, Norway

https://orcid.org/0000-0003-1061-1131

**Andrea Kis** is in the Department of Industrial Engineering & Innovation Sciences, Eindhoven University of Technology, Eindhoven, The Netherlands

https://orcid.org/0000-0002-4345-3814

**Alicja Koperska** is at the Poznan University of Business and Economics, Poznan, Poland

https://orcid.org/0000-0003-2075-7732

**Marie Adrienne Robles Manalili** is at AGHAM Advocates of Science and Technology for the People, Quezon City, Philippines

https://orcid.org/0000-0003-1564-8865

**Adrien Mathy** is at the ULiège Library and the Center of Semiotic and Rhetoric, University of Liege, Liege, Belgium

https://orcid.org/0000-0002-8459-359X

**Christopher J Graham** is at the Royal College of Physicians of Edinburgh (RCPE), Edinburgh, United Kingdom

https://orcid.org/0000-0002-1144-7970

**Anna Hollis** is at the Queen's University of Belfast, Belfast, United Kingdom

https://orcid.org/0009-0006-6006-8498

**Robert M Ross** is in the Department of Philosophy, Macquarie University, Sydney, Australia

https://orcid.org/0000-0001-8711-1675

**Siu Kit Yeung** is in the Department of Psychology, Chinese University of Hong Kong, Hong Kong, China

https://orcid.org/0000-0002-5835-0981

**Veronica Allen** is at the Kapteyn Astronomical Institute, University of Groningen, Groningen, The Netherlands

https://orcid.org/0000-0002-8021-0344

**Flavio Azevedo** is in the Department of Interdisciplinary Social Science, University of Utrecht, Utrecht, The Netherlands

https://orcid.org/0000-0001-9000-8513

**Emily Friedel** is in the Centre for Social and Early Emotional Development and the School of Psychology, Deakin University, Burwood, Australia

https://orcid.org/0009-0001-0917-5398

**Stephanie Fuller** is at Ask Me, I'm an AAC user and is based in the United States

**Vaitsa Giannouli** is at the School of Medicine, Aristotle University of Thessaloniki, Thessaloniki, Greece

https://orcid.org/0000-0003-2176-8986

**Biljana Gjoneska** is at the Macedonian Academy of Sciences and Arts, Skopje, North Macedonia
🆔 https://orcid.org/0000-0003-1200-6672
**Helena Hartmann** is in the Department of Neurology, University Hospital Essen, Essen, Germany
🆔 https://orcid.org/0000-0002-1331-6683
**Max Korbmacher** is at Western Norway University of Applied Sciences, Bergen, Norway
🆔 https://orcid.org/0000-0002-8113-2560
**Mahmoud M Elsherif** is at the University of Birmingham, Birmingham and the University of Leicester, Leicester, United Kingdom
🆔 https://orcid.org/0000-0002-0540-3998
**Alyssa Hillary Zisk** is at the University of Rhode Island, South Kingstown, United States, and Ask Me, I'm an AAC user, United States
🆔 https://orcid.org/0000-0003-2266-4855

*Author contributions:* Mirela Zaneva, Conceptualization, Data curation, Formal analysis, Supervision, Validation, Investigation, Methodology, Writing – original draft, Project administration, Writing – review and editing; Tao Coll-Martín, Investigation, Writing – original draft, Writing – review and editing; Yseult Héjja-Brichard, Investigation, Writing – original draft, Writing – review and editing; Tamara Kalandadze, Formal analysis, Validation, Investigation, Writing – original draft; Andrea Kis, Investigation, Writing – original draft, Writing – review and editing; Alicja Koperska, Investigation, Writing – original draft, Writing – review and editing; Marie Adrienne Robles Manalili, Investigation, Writing – original draft, Writing – review and editing; Adrien Mathy, Investigation, Writing – original draft, Writing – review and editing; Christopher J Graham, Formal analysis, Validation, Investigation, Writing – review and editing; Anna Hollis, Validation, Investigation, Writing – review and editing; Robert M Ross, Formal analysis, Investigation, Writing – review and editing; Siu Kit Yeung, Formal analysis, Investigation, Writing – review and editing; Veronica Allen, Investigation; Flavio Azevedo, Supervision, Project administration, Writing – review and editing; Emily Friedel, Validation, Writing – review and editing; Stephanie Fuller, Investigation, Writing – review and editing; Vaitsa Giannouli, Investigation, Writing – review and editing; Biljana Gjoneska, Investigation, Writing – review and editing; Helena Hartmann, Investigation, Writing – review and editing; Max Korbmacher, Investigation, Writing – review and editing; Mahmoud M Elsherif, Conceptualization, Data curation, Formal analysis, Supervision, Validation, Investigation, Methodology, Writing – original draft, Project administration, Writing – review and editing; Alyssa Hillary Zisk, Formal analysis, Supervision, Validation, Investigation, Methodology, Project administration, Writing – review and editing

*Competing interests:* The authors declare that no competing interests exist.

## Funding

| Funder | Grant reference number | Author |
|---|---|---|
| Leverhulme Trust | | Mahmoud M Elsherif |
| John Templeton Foundation | 62631 | Robert M Ross |
| NWO Veni | | Veronica Allen |
| Spanish Ministry of Economy and Competiveness | PID2020-114790GB-I00 | Tao Coll-Martín |
| University of Granada | Contrato Puente-Plan Propio UGR | Tao Coll-Martín |

The funders had no role in study design, data collection and interpretation, or the decision to submit the work for publication.

**Decision letter and Author response**
Decision letter https://doi.org/10.7554/eLife.102467.sa1
Author response https://doi.org/10.7554/eLife.102467.sa2

# Additional files

## Supplementary files
• MDAR checklist

## Data availability
No data were generated for this work.

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

## Appendix 1

Short, non-exhaustive summaries of significant contributions by the articles in our reading list.

| Text | Contributions |
|---|---|
| **History of neurodiversity** | |
| Botha M, Chapman R, Giwa Onaiwu M, Kapp SK, Stannard Ashley A, Walker N. 2024. The neurodiversity concept was developed collectively: An overdue correction on the origins of neurodiversity theory. *Autism* **28**:1591–1594. | Letter authored by an international group of autistic scholars, powerful collaborative work<br>Corrects erroneous beliefs about the origins of the concept of neurodiversity<br>Highlights the power of community discussions and community theorising |
| Sinclair J. 1993. Don't mourn for us. *Autism Network International: Our Voice Newsletter* Issue 3; pages 20–22. | Challenges deficit-based models of autism, argues that autism isn't a tragedy<br>Advocates for acceptance, particularly from parents and society<br>Encourages a reframing of the parent-child relationship, and building a positive relationship with one's child, embracing who they are as an individual |
| **How do we think about neurodiversity?** | |
| Dwyer P. 2022. The neurodiversity approach(es): What are they and what do they mean for researchers? *Human Development* **66**:73–92. | Clear definitions of key terms in neurodiversity<br>Different approaches are well situated and intelligible for broad audience<br>Implications for developmental psychology are presented |
| Constantino CD. 2018. What can stutterers learn from the neurodiversity movement? *Seminars in Speech and Language* **39**:382–396. | Neurodiversity has placed a large focus on autism and ADHD, so this is an especially important perspective on how the neurodiversity movement can benefit stutterers<br>Clear communication on how therapy objectives can correspond to people's lives<br>Focus on mental health and wellbeing as goals |
| **The importance of lived experience** | |
| Johnson RM. 2023. Dyslexia is not a gift, but it is not that simple. *Infant and Child Development* **32**:e2454. | An important perspective on dyslexia, including a survey of different conceptualizations of dyslexia<br>Well contextualized discussion on the 'dyslexia gift' debate<br>Places importance on research partnership and collaboration |
| van Gorp R. 2022. My journey and the value of a community where neurodiversity is celebrated. *Scope Contemporary Research Topics: Learning and Teaching* **11**:42–49. | Powerful lived experience report (Irlen Syndrome and dyslexia)<br>Discusses issues around deciding to disclose being neurodivergent<br>Argues for and illustrates benefits of community participation |
| **A neurodiversity paradigm for autism science** | |
| Pellicano E, den Houting J. 2022. Annual Research Review: Shifting from 'normal science' to neurodiversity in autism science. *Journal of Child Psychology and Psychiatry* **63**:381–396. | Powerful argument for embracing a neurodiversity perspective in autism science<br>High educational value, providing clear overview of conventional medical paradigm and critiques of it<br>Well-presented definitions and core perspectives from the neurodiversity paradigm |
| Botha M, Hanlon J, Williams GL. 2023. Does language matter? Identity-first versus person-first language use in autism research: A response to Vivanti. *Journal of Autism and Developmental Disorders* **53**:870–878. | Strong arguments presented regarding how language use has important consequences for neurodivergent people, specifically for autistic people<br>Valuable educational resources, discusses differences between identity-first and person-first language use with clear examples<br>Clear illustrations of pitfall of superficial engagement with autistic scholarship |

*Continued*

| Text | Contributions |
|------|---------------|
| **Beyond deficit views of ADHD** | |
| Sonuga-Barke EJ. 2023. Paradigm 'flipping' to reinvigorate translational science: Outlining a neurodevelopmental science framework from a 'neurodiversity' perspective. *Journal of Child Psychology and Psychiatry* **64**:1405–1408. | Challenges deficit-based view in neurodevelopmental fields of autism and ADHD<br>Encourages participation of neurodivergent people in the full scientific process<br>Offers concrete examples of participatory practices |
| Tamir T. 2023. Being Neurodivergent in Academia: Working with my brain and not against it. *eLife* **12**:e95068. | A powerful self-report of a neurodivergent person's lived experiences in navigating their academic and personal life<br>An informative and personal account of coping-strategies<br>Offers intersectional lens, particularly regarding mental health and cultural background |
| **Expanding the scope of neurodiversity: diverse neurotypes and experiences** | |
| Green A. et al. 2020. Teaching and researching with a mental health diagnosis: Practices and perspectives on academic ableism. *Rhetoric of Health & Medicine* **3**: Issue 2, article 1. | Informative piece addressing the overlap between neurodiversity and mental health<br>Highlights diagnoses that are not typically considered in neurodiversity (e.g., bipolar and personality disorders)<br>Discusses realities and impacts of academic ableism |
| Elsherif MM, Wheeldon LR, Frisson S. 2021. Do dyslexia and stuttering share a processing deficit? *Journal of Fluency Disorders* **67**:105827. | Focuses on an important, understudied topic within neurodiversity<br>Challenges previous research which had ignored the co-occurrence between dyslexia and stuttering<br>Shows commonalities between dyslexia and stuttering |
| **Anti-ableism** | |
| Natri HM, et al. 2023. Anti-ableist language is fully compatible with high-quality autism research: Response to Singer et al.(2023). *Autism Research* **16**:673–676. | Offers a powerful argument that anti-ableism and neurodiversity frameworks do not conflict with scientific accuracy or quality<br>Calls into question several claims from those arguing ableist language are needed<br>The text is a letter, authored by a group of autistic researchers, scholars, clinicians, self-advocates, and showcases the importance collaborative work |
| Hamilton LG, Petty S. 2023. Compassionate pedagogy for neurodiversity in higher education: A conceptual analysis. *Frontiers in Psychology* **14**:1093290. | Focuses on the need for educators' empathy, rather than seeing neurodivergent students as the problem<br>Concrete illustrations of building positive learning environments<br>Makes case for neurodiversity-friendly higher education environments |
| **The need for robust theory and methods** | |
| Gernsbacher MA, Yergeau M. 2019. Empirical failures of the claim that autistic people lack a theory of mind. *Archives of Scientific Psychology* **7**:102. | Challenges the pervasive claim that autistic people lack theory of mind<br>Offers a critical look at the literature supporting this claim and suggests numerous methodological faults<br>Concretely illustrates not only how shaky the foundations of this claim are, but also how societally harmful it has been |
| Cheng Y, Tekola B, Balasubramanian A, Crane L, Leadbitter K. 2023. Neurodiversity and community-led rights-based movements: Barriers and opportunities for global research partnerships. *Autism* **27**:573–577. | A very important piece regarding global perspectives of the neurodiversity paradigm and movement<br>The vast majority of neurodiversity articles derive from WEIRD countries, and so many global perspectives are ignored<br>Concrete discussions of barriers and opportunities |

*Continued on next page*

*Continued*

| Text | Contributions |
|---|---|
| **Integration with open and participatory work** | |
| Gourdon-Kanhukamwe A. et al. 2023. Opening up understanding of neurodiversity: A call for applying participatory and open scholarship practices. *The Cognitive Psychology Bulletin* **8**:23–27. | Offers a poignant call to redress the power imbalances regarding inclusion in neurodiversity<br>Particular focus on research, where the practices of open scholarship and participatory work are offered as solutions<br>Accessible to wider audience, well structured |
| Heraty S. et al. 2023. Bridge-building between communities: Imagining the future of biomedical autism research. *Cell* **186**:3747–3752. | Focuses on tangible ways to strengthen and include neurodiversity perspective in biomedical science<br>Particularly important given the context of most biomedical research is still being carried with a deficit-oriented perspective<br>Includes strategies to minimize risks and harm, advocates for the active inclusion of autistic people in the research process |

