## [Decision Letter]

**Decision letter after peer review:**

Thank you for submitting your article "An Annotated Introductory Reading List for Neurodiversity" to *eLife* for consideration as a Feature Article. Your article has been reviewed by three peer reviewers, and the evaluation has been overseen by Peter Rodgers of *eLife*. The following individuals involved in review of your submission have agreed to reveal their identity: Kristen Bottema-Beutel (reviewer 1); Hanna Bertilsdotter-Rosqvist (reviewer 3).

The reviewers and editors have discussed the reviews and we have drafted this decision letter to help you prepare a revised submission.

General assessments

*Reviewer #1:*

The authors have provided an informative and comprehensive annotated reading list to introduce scholars and laypeople to key concepts and areas of inquiry within neurodiversity scholarship. The annotated list clarifies complex issues in a way that will be accessible to scholars in a wide variety of disciplines as well as non-scholars.

Overall, this annotated list was a pleasure to read – but I do have a few recommendations for the authors to improve upon what is already an excellent manuscript.*Reviewer #2:*

Thank you for the opportunity to review this article. I thoroughly enjoyed reading it, as it presents a comprehensive reading list crucial for understanding neurodiversity, particularly for those less familiar with the concept. I appreciate the collaborative efforts between neurodivergent and neurotypical researchers, which enrich the discussions and provide balanced perspectives on various relevant topics. Notably, the article offers clear definitions of terms relevant to neurodiversity, addressing inconsistencies in current usage and establishing a solid foundation for future research to reference. The historical context and related frameworks highlighted, which have at times been overlooked or mistaken, are also valuable.

Overall, I believe this article will serve as a significant resource for a wide audience.

*Reviewer #3:*

I may be hesitant to this establishing of a "canon" – as it is always situated/constructed, but I also acknowledge that this bibliography is most helpful for newcomers to the field (and I love the methods/the ways this big task has been accomplished).

Summary

The authors have provided an informative and comprehensive annotated reading list to introduce scholars and laypeople to key concepts and areas of inquiry within neurodiversity scholarship. The fact that the article is the result of a collaborative effort involving both neurodivergent and neurotypical researchers enrich the discussions and provides balanced perspectives on various relevant topics. Notably, the article offers clear definitions of terms relevant to neurodiversity, addressing inconsistencies in current usage and establishing a solid foundation for future research to reference. However, there are a number of concerns that need to be addressed, as outlined below.

Essential revisions

1) I recommend clarifying terminology sooner in the manuscript, and using terms consistently. The authors open the paper with the term "neurodiversity", but seem to mean the "neurodiversity paradigm" or "neurodiversity framework" for understanding disability. As the authors clarify in the second paragraph, if "neurodiversity" refers to the biological fact that of neurocognitive variation, it should not be used as a term to describe a conceptual framework.

2) In addition to the helpful definitions of neurodiversity and neurodivergence, it might be beneficial to clarify the distinctions between the two and provide guidance on their usage. This could help reduce confusion and uncertainty, thereby mitigating potential tensions and misunderstandings in the field.

3) The authors should clarify to whom the open call for reading recommendations was made and offer a general sense of who provided recommendations (scholars? laypeople? a mix of both?). I see the details about FORRT in the supplemental materials, but I am not familiar with this platform and am not sure what audiences would have received this call.

4) There do seem to be some seminal texts missing from the annotated list- the list seems to favor briefer articles published in academic journals (Jim Sinclair's piece being an exception). The seminal texts that came to mind are:

Milton, D., Ridout, S., Murray, D., Martin, N., and Mills, R. (2020). The Neurodiversity Reader: Exploring concepts, lived experiences and implications for practice. Hove, UK: Pavilion.

Kapp, S.K. (Ed.) (2020). Autistic community and the neurodiversity movement: stories from the frontline. Palgrave Macmillan (this does appear in the reference list, but is not a selected text)

Walker, N. (2021). Neuroqueer heresies. Autonomous press.

Yergeau, M. (2018). Authoring autism: On rhetoric and neurological queerness. Duke University Press.

I am not necessarily suggesting that these texts need to be included in the annotated list, but the authors may want to make specific mention of them in the text as additional reading, and perhaps clarify that the annotated list favored shorter pieces as opposed to book-length theses or anthologies.

5) One additional omission is issues related to intersectionality, and who is traditionally included in neurodiversity activism and scholarship. There have been a few recent papers published in the journal Autism (one published very recently, so it would have not been available to the authors at the time of their writing) that the authors may want to mention as raising important issues (even if they are not selected for the list):

Cheng, Y., Tekola, B., Balasubramanian, A., Crane, L., and Leadbitter, K. (2023). Neurodiversity and community-led rights-based movements: Barriers and opportunities for global research partnerships. Autism, 27(3), 573-577.

Nair, V. K., Farah, W., and Boveda, M. (2024). Is neurodiversity a Global Northern White paradigm? Autism, 13623613241280835.

6) I have particular concerns regarding the structure of the main theme "Current Topics". The subthemes do not seem to effectively correspond to this overarching theme and would benefit from further categorisation. For instance, the subthemes "Autism" and "ADHD" are not sufficiently informative on their own, and their content could be further extracted and synthesised. The discussion on medical versus neurodiversity-affirming models, for example, emerges as a common current topic across the "Autism", "ADHD", and "Beyond Autism and ADHD" subthemes. Other examples of potential subthemes could be language and identity, the need to expand the scope of neurodiversity, mental health, given the current annotations available. Relatedly, the personal journey of an academic with a diagnosis of depression and ADHD under "ADHD" might be more readily integrated into the section on "The importance of lived experiences".

Overall, the presentation currently feels somewhat disorganised and lack the synthesis needed to effectively highlight current topics in the field. It would also be beneficial to explore how this theme can be expanded, even if it involves including additional items on the reading list, to reflect the broader contemporary discussions relevant to neurodiversity.

[note from the Editor: It is not essential that you revise the subthemes under Current Topics. However, if you retain the current subthemes, please discuss other subthemes that could have been used]

[7] Something I was a bit disappointed in finding, was the relative dominance of scientific papers, as I expect it is much harder for newcomers to the field to find the important community theorising published in social media (blogs/vlogs). So I would have appreciated something more from the blogosphere + suggesting perhaps another section focusing on community theorising, and discussing academic versus community theorising a bit. And when I say this, I also acknowledge that my personal favourites and important inspirations among the community theorists was not included, so I am mindful that this bibliography is just a selection which the authors are clearly stating.

---

## [Author Response]

Essential revisions1) I recommend clarifying terminology sooner in the manuscript, and using terms consistently. The authors open the paper with the term "neurodiversity", but seem to mean the "neurodiversity paradigm" or "neurodiversity framework" for understanding disability. As the authors clarify in the second paragraph, if "neurodiversity" refers to the biological fact that of neurocognitive variation, it should not be used as a term to describe a conceptual framework.

Thank you, this is helpful.

In our revision, we have first redrafted the Introduction such that we open with the term neurodiversity paradigm:

'The neurodiversity paradigm can be seen as a shift in thinking that embraces the diversity of minds, brains, and neurocognition and affirms variation as natural and valuable (Pellicano and den Houting, 2022; Walker, 2021).'

We have also reworked the sections so that we clarify terminology at an earlier stage in the Introduction. Specifically, we have revised the ordering and presentation of our previous paragraphs 1 and 2.

'Like other paradigm shifts, this change in perspective can be challenging to fully understand in the context of education, research, and social thinking. Here, the issue is further compounded as the term ‘neurodiversity’ has broad scope. Numerous definitions of neurodiversity as a movement, a research field, and a framework or paradigm exist (Dwyer, 2022; Milton et al., 2020). These three aspects can be distinguished in the following way: (1) the neurodiversity movement encompasses social, advocacy, and political movements advocating for the rights, inclusion, and acceptance of neurodivergent people; (2) the neurodiversity research field is a largely academic field studying, for instance, psychological and social aspects of neurodiversity; (3) the neurodiversity paradigm or framework is a conceptual framework that at its core challenges medical or deficit-based views of neurodiversity and instead asserts that neurocognitive differences should be seen as natural variations (see Table 1). It is important to acknowledge that neurodiversity is inherently interdisciplinary, and these three aspects can overlap and intersect, and jointly contribute to a more holistic understanding of neurodiversity. For instance, research and activism can intersect in areas such as disability rights, mental health advocacy, social justice, and equity, diversity and inclusion efforts in education and in the workforce (Clouder et al., 2020; Dwyer, 2022; Manalili et al., 2023). Unless specifically discussing one of these aspects, here we generally adopt the term ‘neurodiversity paradigm’ as a broader idea, encompassing ways of thinking applicable to both the neurodiversity movement and the research field.'

Further, we have gone through the full manuscript and ensured we have consistency of terminology use in line with the definitions we offer in the Introduction.

2) In addition to the helpful definitions of neurodiversity and neurodivergence, it might be beneficial to clarify the distinctions between the two and provide guidance on their usage. This could help reduce confusion and uncertainty, thereby mitigating potential tensions and misunderstandings in the field.

Thank you for this suggestion. This is very useful guidance for us, as it is our goal to provide an introductory resource that is indeed helpful and clear. To this end, we have addressed your comment by creating a new table (Table 1), which contains definitions of neurodiversity and neurodivergence. We refer to this table in the Introduction. We have followed the spirit of your comment to reduce confusion and uncertainty, and extended it to a few other terms that might be less clear to a diverse readership. The table contains a brief description of each term as well as further notes. The ‘Further notes’ column uses the term (and so provides a usage example) and offers further clarification and referencing.

3) The authors should clarify to whom the open call for reading recommendations was made and offer a general sense of who provided recommendations (scholars? laypeople? a mix of both?). I see the details about FORRT in the supplemental materials, but I am not familiar with this platform and am not sure what audiences would have received this call.

Thank you, we have addressed this by providing further detail in the manuscript and supplement.

In the manuscript we have added:

'The open call was primarily shared through the Framework for Open and Reproducible Research Training (FORRT) platform, an online collaborative community focused on promoting open scholarship, as well as advancing research and education about neurodiversity.'

In the supplement we have added:

'We believe FORRT reaches a fairly diverse audience of people working in and outside of academia, interested in open science and neurodiversity. Some of our collaborators were already involved in other FORRT projects, whereas others were familiar with FORRT but not involved in this community; some collaborators also joined after hearing about our open call from their own networks.'

4) There do seem to be some seminal texts missing from the annotated list- the list seems to favor briefer articles published in academic journals (Jim Sinclair's piece being an exception). The seminal texts that came to mind are:Milton, D., Ridout, S., Murray, D., Martin, N., and Mills, R. (2020). The Neurodiversity Reader: Exploring concepts, lived experiences and implications for practice. Hove, UK: Pavilion.Kapp, S.K. (Ed.) (2020). Autistic community and the neurodiversity movement: stories from the frontline. Palgrave Macmillan (this does appear in the reference list, but is not a selected text)Walker, N. (2021). Neuroqueer heresies. Autonomous press.Yergeau, M. (2018). Authoring autism: On rhetoric and neurological queerness. Duke University Press.I am not necessarily suggesting that these texts need to be included in the annotated list, but the authors may want to make specific mention of them in the text as additional reading, and perhaps clarify that the annotated list favored shorter pieces as opposed to book-length theses or anthologies.

Thank you, this is helpful guidance to receive, and wonderful texts to receive as further recommendations. We had included references to both Kapp (2020) and a section of Walker (2021) but on reflection, as well as examining the other two suggested references more carefully, we agree these are important to more strongly highlight. Indeed, we were initially somewhat hesitant to recommend books and lengthier texts, as we worried the length might discourage newcomers (and have now added more detail in the Methods section in the Supplement regarding accessibility of texts). Equally, some of these texts contain very powerful chapters or sections relating to at times quite distinct topics, and we wondered to what extent we could appropriately annotate the full breadth of issues covered in these texts. For instance, The Neurodiversity Reader covers important topics ranging on diverse issues such as perspectives on autism, neurodiversity beyond autism, female neurodiversity, employment and university experiences, a literature review on person-centered counseling etc. With all this in mind, we have decided not to directly recommend these texts in our themes but rather, we have opted to take a three step approach to better showcase these important texts. First, we better highlight these in the manuscript, second, we have added them as direct recommendations in our new ‘Further readings’ section, and third, we have highlighted them in our References section.

To expand on the first point, we have referenced the texts throughout the manuscript, to better highlight immediate areas of connection and relevance:

Milton et al. (2020): We reference this both in our Introduction, in the context of our discussion of the concepts of neurodiversity, as well as in the theme on the importance of lived experience.Kapp (2020): We include this in our discussion of the history of neurodiversity theme, as well as in the theme on autism.Walker (2021): We have more directly highlighted Walker (2021) in our new Table 1, where we both direct readers to this text in the table description, as well as in the table itself as a direct reference.Yergeau (2018): We reference this in our theme on the history of neurodiversity.

On the second point, in our new ‘Further readings’ section, we have added a mention to all these texts.

'We hope the themes discussed here spark an interest in neurodiversity. The current list of themes is not intended as a canonical or definitive organization but is only one of many possible ways to learn more about neurodiversity. As such, we want to offer suggestions for further key readings, as well as other potential themes or topics of interest. Readers interested in extended introductions about the neurodiversity paradigm, movement, and research field, may wish to consider The Neurodiversity Reader (Milton et al., 2020). Detailed accounts about the autistic community, its early development, and key figures should consult Kapp (2020). For deeper theoretical engagement with neurodiversity, as well as critiques to medicalized views and harmful societal viewpoints should see Walker (2021) and Yergeau (2018).'

On the third point, in our References section we have added an * symbol to denote particular texts we recommend for readers interested in further reading. We have marked all four suggested works for further reading there as well.

5) One additional omission is issues related to intersectionality, and who is traditionally included in neurodiversity activism and scholarship. There have been a few recent papers published in the journal Autism (one published very recently, so it would have not been available to the authors at the time of their writing) that the authors may want to mention as raising important issues (even if they are not selected for the list):Cheng, Y., Tekola, B., Balasubramanian, A., Crane, L., and Leadbitter, K. (2023). Neurodiversity and community-led rights-based movements: Barriers and opportunities for global research partnerships. Autism, 27(3), 573-577.Nair, V. K., Farah, W., and Boveda, M. (2024). Is neurodiversity a Global Northern White paradigm? Autism, 13623613241280835.

We appreciate the reviewer's thoughtful comment highlighting the critical importance of intersectionality in neurodiversity activism and scholarship – this is something we agree with very much. In our original draft, we had alluded to the importance of intersectionality broadly but on reflection, we see how this was not well fleshed out throughout the manuscript. To this end, we have incorporated both of the references here and revised our manuscript to better clarify the importance of recognizing intersectionality.

Specifically, we highlight Cheng et al. as our recommendation in the theme 'The need for robust theory and methods.' We also use Nair et al. in the same paragraph, alongside a further new reference of Giwa Onaiwu’s (2020) powerful words about her experience as a Black autistic mother to Black autistic children and the importance of racial diversity. This section now reads:

'Further important aspects for developing robust methods for studying neurodiversity include asking useful research questions with relevance to neurodivergent people’s lives and needs (see next theme), questioning who gets to be included in conversations and work on neurodiversity, and understanding neurodiversity as a global, rather than solely western area of research and activism. Historically, the neurodiversity movement has been driven largely by English-speaking White autistic people, primarily from countries in the Global North and there is still a great need to recognize the issue of intersectionality in terms of whose voices have been included in neurodiversity activism and scholarship and whose have been excluded (for instance, on the need for greater racial diversity in autism research, see Giwa Onaiwu, 2020). Indeed, as some have warned, solidifying the homogenization of neurodiversity as a White and western movement could undermine the social justice and emancipatory goals of the movement (Nair, Farah, and Boveda, 2024). Drawing on their collective experiences in Ethiopia, India, and Hong Kong, Cheng and colleagues (2023) can help readers think critically about the issues of intersectionality and inclusion, by discussing a variety of sociocultural and political conditions specific to Asian and African neurodiversity efforts. The authors argye that the neurodiversity movement shares fundamental goals with decolonization agendas such as dismantling what, at times, to some may have seemed as 'objective' scientific efforts that ultimately disparage the truths, knowledge, and priorities of lived experiences (e.g., claims that autistic people lack theory of mind). In this light, decolonizing knowledge production, respecting local theoretical frameworks, indigenous knowledge, and fostering community-led science could be important tools for a more robust study of neurodiversity that does not dehumanize neurodivergent people.'

[6] I have particular concerns regarding the structure of the main theme "Current Topics". The subthemes do not seem to effectively correspond to this overarching theme and would benefit from further categorisation. For instance, the subthemes "Autism" and "ADHD" are not sufficiently informative on their own, and their content could be further extracted and synthesised. The discussion on medical versus neurodiversity-affirming models, for example, emerges as a common current topic across the "Autism", "ADHD", and "Beyond Autism and ADHD" subthemes. Other examples of potential subthemes could be language and identity, the need to expand the scope of neurodiversity, mental health, given the current annotations available. Relatedly, the personal journey of an academic with a diagnosis of depression and ADHD under "ADHD" might be more readily integrated into the section on "The importance of lived experiences".Overall, the presentation currently feels somewhat disorganised and lack the synthesis needed to effectively highlight current topics in the field. It would also be beneficial to explore how this theme can be expanded, even if it involves including additional items on the reading list, to reflect the broader contemporary discussions relevant to neurodiversity.[note from the Editor: It is not essential that you revise the subthemes under Current Topics. However, if you retain the current subthemes, please discuss other subthemes that could have been used]

This is valuable feedback, thank you. We understand that the Reviewer’s main concerns here are related to the need to improve how informative the themes are, as well as the overall organization and presentation. We have taken a number of steps to improve these.

First, we have removed the bigger theme headings ('What is neurodiversity?', 'Current topics', 'Improving the field'). As the reviewer points out, there are commonalities shared between some of our themes (such as around language, identity, challenging medical models etc.). From that perspective, we considered that some of these bigger headings implied clear-cut divisions that could be unhelpful (whereas, for instance, some of the current topics relate to an understanding of what neurodiversity is, and some of the topics under ‘Improving the field’ are both current topics and helpful for understanding views of what neurodiversity is).

Second, we have worked to improve specificity of each theme’s content by revising some of the titles.

'Autism': This theme is now called 'A neurodiversity paradigm for autism science***'***. We believe this now better summarizes the content of the theme, as both of the papers we highlight argue for a neurodiversity paradigm for autism science, albeit with different specific examples.'ADHD': This theme is now called 'Beyond deficit views of ADHD'. As above, we trust this is now more specific about the content of the theme.'Beyond Autism and ADHD': This is now called 'Expanding the scope of neurodiversity: diverse neurotypes and experiences'.Third, we have revised our text in order to highlight commonalities between the themes, and better present the content of the themes. In the Introduction, we have expressed this in order to better guide the reader:

'Rather than being completely discrete categories, the themes we present often raise shared discussion points, albeit from different perspectives and with different examples. For instance, commonalities emerge regarding the importance of neurodiversity affirming models over deficit-based views, the use of appropriate language, the need to expand definitions of neurodiversity (for instance, to be inclusive of mental health), and the need for inclusion and centering of lived experience.'

Further, we understand the Reviewer suggests moving the van Gorp (2022) text to the section on lived experience. This was a popular piece and indeed was considered a good fit for both the themes on lived experience and ADHD. After discussions, we decided that we already had many texts highlighting autism and ADHD (in part reflecting the field itself) and that if we highlighted a text on ADHD in the lived experience theme, this would perhaps become even more disproportionate. We see value in using the section on lived experiences to highlight other experiences beyond autism and ADHD as in our view, until recently, such experiences have been less often focused on and acknowledged. Similarly, we also thought it would be beneficial to have a report of one’s lived experience in the ADHD section in regards to the same core issue brought up by the Reviewer in the below comment 7, that is: who gets to be part of conversations about neurodiversity? Although comment 7 focuses on a specific medium as an example (blogs), we believe it also points to a question of who the enunciator is, with which we agree: we do not wish to place too much (or exclusive) emphasis on 'expert' speech or discourse (e.g., academic reports, experiments or interventions).

Lastly, in response to the Editor’s note, we have discussed other subthemes that could have been used in our new section ‘Further readings’:

'It is important to recognize the plurality of important topics in neurodiversity and the inherent subjectivity in thematic categorization. For instance, many of the papers we synthesized here could have been thematically organized in different ways, such as focusing more on the evolution of the neurodiversity paradigm over time, different global and cultural perspectives on neurodiversity, the importance of intersectionality in shaping neurodivergent experiences, better understanding stigma, ableism, and language, as well as charting neurodiversity across the lifespan, or understanding how lived experience not only can drive academic research and theory, but can itself be a theoretical contribution.'

7) Something I was a bit disappointed in finding, was the relative dominance of scientific papers, as I expect it is much harder for newcomers to the field to find the important community theorising published in social media (blogs/vlogs). So I would have appreciated something more from the blogosphere + suggesting perhaps another section focusing on community theorising, and discussing academic versus community theorising a bit. And when I say this, I also acknowledge that my personal favourites and important inspirations among the community theorists was not included, so I am mindful that this bibliography is just a selection which the authors are clearly stating.

Thank you, this is a really important comment. We feel strongly about the inclusion of literature outside the scientific field, and this is why we set our project from the start to be inclusive of non-academic texts. The lead authors also tried to encourage the team to further consider blogs, opinions, manifestos etc. At the same time, we recognize and agree with the Reviewer that we have predominantly highlighted academic texts. In our work process, this emerged as a trend given methodological and practical considerations, chiefly that a certain degree of homogeneity between the chosen texts would perhaps be helpful to newcomers (for instance, most academic articles tend to have a common structure), as well as the fact that most academic articles meet certain standards of evaluation unlike some of the speech we find on blogs, which may often require further forms of specific contextualization.

Nevertheless, we strongly agree with the Reviewer on the importance of highlighting such work further. We have strived to do this in our new ‘Further readings’ sections. We have first reminded readers that this is not a 'canonical' or definitive list (hopefully also addressing Reviewer 3’s comment in the General Assessments):

'Our list of themes is not intended as a canonical or definitive categorization but is only one of many possible ways to learn more about neurodiversity. […] It is important to recognize the plurality of important topics in neurodiversity and the inherent subjectivity in thematic categorization.'

Then, we have discussed the importance of non-academic work and community theorizing, and have offered two starting points as concrete examples:

'Indeed, as online forums and discussions show, the term neurodiversity was collectively developed (Botha et al., 2024; see the *History of neurodiversity* theme). This is only one of many examples of the power of community discussion and community theorizing (Zisk, 2024). With this in mind, we also wish to encourage interested readers to consider blogs. As one example, we recommend Patrick Dwyer’s blog *Autistic Scholar* with rich discussions, informed by both academic research and lived experience on contemporary topics such as, for instance, the double empathy problem (Dwyer 2024). Blogs can also chart the evolution and spread of ideas. For instance, Mel Baggs’ important discussion on the history and importance of the concept of neurodivergent ‘cousins’, that is, people who share common communication patterns or social characteristics without necessarily sharing the same neurotype (Baggs, 2016).'

Finally, we also point readers to an existing annotated reading list that focuses on critical autism studies specifically beyond academia (Zisk, 2023):

'We also direct interested readers to consult the reading list by Zisk (2023) on critical autism studies beyond academia.'